# Identification of Vegetation Coverage Variation and Quantitative the Impact of Environmental Factors on Its Spatial Distribution in the Pisha Sandstone Area

Lu Jia [1], Kunxia Yu [1,*], Zhanbin Li [1], Zongping Ren [1,2], Hongtao Li [3] and Peng Li [1,2]

1 State Key Laboratory of Eco-Hydraulics in Northwest Arid Region of China, Xi'an University of Technology, Xi'an 710048, China
2 Key Laboratory of National Forestry and Grassland Administration on Ecological Hydrology and Disaster Prevention in Arid Regions, Xi'an 710048, China
3 Northwest Surveying, Planning Institute of National Forestry and Grassland Administration, Xi'an 710048, China
* Correspondence: yukunxia@126.com

**Abstract:** Over the past few decades, global vegetation cover has obviously changed, particularly in the Loess Plateau, due to vegetation restoration projects in China. This study focuses on the Pisha Sandstone area (PSA) and uses various statistical analysis methods to study the spatiotemporal changes in vegetation coverage (VEC) at different time scales. The effects of topographical and climatic factors on VEC were also quantitatively evaluated using the GeoDetector in the spatial distribution. The results of the study confirm that, on an annual scale, the area with a significant increase in VEC has reached 63.89% ($p < 0.05$). Change points were diagnosed to have occurred mainly between 2002 and 2012 at different time scales, with the percentage of significant change points in VEC accounting for more than 20% from April to October ($p < 0.05$). Temporal and spatial changes in precipitation mainly caused VEC changes. In 45.35% of the region, precipitation was significantly and positively correlated with VEC at an annual scale ($p < 0.05$). Moreover, VEC was most conducive to growth and increase at 1050–1500 m above sea level and 0–21° slope, respectively. In most areas, there was an enhanced interaction relationship between various factors on VEC. Converting farmland to forests in suitable areas, selecting appropriate tree species, and improving soil is conducive to ecological restoration in the PSA in the future.

**Keywords:** Loess Plateau; Pisha Sandstone area; vegetation coverage; environmental factors

## 1. Introduction

Vegetation cannot be replaced to reflect the energy exchange, material conversion, and carbon cycle [1–6]. Vegetation is an important component of carbon sinks in terrestrial ecosystems, and studying changes in VEC is important for understanding the carbon cycle process [3,7,8]. At least 20% of the land surface is covered by vegetation, which is highly sensitive to climate change and has become an indicator [7,9–15]. Moreover, the growth of vegetation can inhibit regional soil erosion to a certain extent [16–18]. Therefore, monitoring and researching regional VEC changes is essential for understanding and managing soil erosion and changes in the terrestrial ecosystem's carbon sink in a changing environment [19–21].

As a remote sensing product, the normalized vegetation index (NDVI) can describe the VEC status. It has the advantage of covering a wide area and is often used in studies to evaluate the regional VEC status [22–30]. The study of VEC changes based on NDVI has yielded abundant research results. Vegetation changes around the world have been studied and monitored by many scholars. On a global scale, vegetation is turning green against the backdrop of climate warming [4]. For example, the VEC shows a slight increasing trend

in North China [31]. Jingyun et al. also showed that VEC in most parts of China displays an increasing trend [32]. There is an obvious upward trend in VEC in the mid-latitudes of the northern hemisphere [24,33]. The research shows that the trend of VEC in Eurasia is consistent [28,34]. Xu et al.'s research indicates that VEC is continuously improving in 83.34% of the regions in China [30]. VEC changes are mainly affected by natural factors and human activities [4,28,30]. Climate factors mainly include precipitation, temperature, wind speed and so on. In arid and semi-arid regions, precipitation and temperature are the main climatic factors affecting VEC change [4]. At the same time, topographical factors also profoundly affect VEC changes [19,30,35]. Ecological restoration, urbanization, and agricultural production activities are all important human activity factors that cause VEC changes, and urbanization can destroy VEC and cause ecosystem shrinkage through land use analysis [4,6]. The changing trend of VEC has obvious heterogeneity, which is closely related to climate and regional characteristics [7]. However, these studies are very general and only use methods such as correlation analysis or regression equations to roughly discuss the impact of various factors on VEC. Although some scholars have proposed geographical detectors to analyze the spatial heterogeneity of geographical variables, most of this analysis is carried out on the entire study area [36], which cannot spatially reflect the complex effects of various variables on the research object. Therefore, it is very meaningful to quantitatively evaluate the impact of various environmental factors on VEC in spatial distribution, especially the interactive relationship.

As a region seriously affected by soil erosion, the Chinese government has implemented large-scale soil and water conservation projects as well as a policy of converting farmland back to forests in the Loess Plateau [37,38]. After decades of effort, the VEC has significantly improved [39]. The water and sediment of the watershed have been greatly reduced. The PSA, as an ecologically fragile area and an important source of sediment in the Yellow River, has drawn the attention of many scholars due to its special geological and climatic conditions. The PSA has extremely poor climate conditions, including drought and little rainfall, poor soil texture, and a lack of organic matter. The soil is more prone to erosion after precipitation, and vegetation growth in this area is more challenging. While many studies have focused on the relationship between climate change and VEC, there are still few research results on how various factors interact with VEC change in spatial distribution in extremely arid areas, particularly in the PSA. Understanding this issue is of great practical significance for accurately arranging soil and water conservation measures in spatial distribution and achieving ecological protection and high-quality development in the Yellow River basin.

The main purposes of this study are: (1) to study the spatiotemporal variation of VEC; (2) to describe the effect of topographical and climate factors on VEC; and (3) to quantify the influence of various factors on the spatial distribution of VEC.

## 2. Materials and Methods

### 2.1. Study Area and Data

The PSA of the Loess Plateau is located in Inner Mongolia Autonomous Region, Shaanxi Province, and the Shanxi Province in China [40,41]. The altitude ranges from 803 to 1628 m, and the terrain is higher in the northwest and lower in the southeast. The main water systems include the Kuye River, Huangfu River, Gushan River, Qingshui River, and other first-class tributaries of the Yellow River. The climate is classified as an arid and semi-arid continental monsoon climate, with an average annual temperature of 6–9 °C, an average annual precipitation of 280–400 mm, an average annual evapotranspiration of 2200–2750 mm, an average annual wind speed of 2–4 m/s, and a maximum wind speed of 28 m/s. The vegetation in the area is sparse, and the natural vegetation is mainly composed of low plants such as Gramineae, Compositae, Leguminosae, Chenopodiaceae, etc. The artificial vegetation mainly includes sea buckthorn, caragana, Pinus tabulaeformis, etc. The main soil types include yellow cotton soil, chestnut soil, aeolian sand soil, coarse bone soil, and newly accumulated soil, etc. [40]. For this area, the flood season is from June

to October. The Pisha sandstone is a type of fluvial clastic deposition sandstone formed in the Jurassic, Triassic and Cretaceous periods [42]. It consists of sandstone, mudstone, and arenaceous. The area is approximately 16,700 km$^2$, with ravines crisscrossing. The PSA is mainly divided into three sub-areas, namely the bare sandstone sub-area (BSA), the sand-covered sub-area (SASA), and the soil-covered sub-area (SOSA). The areas of the three sub-areas are 8432.40, 3709.18, and 4543.89 km$^2$, respectively (Figure 1).

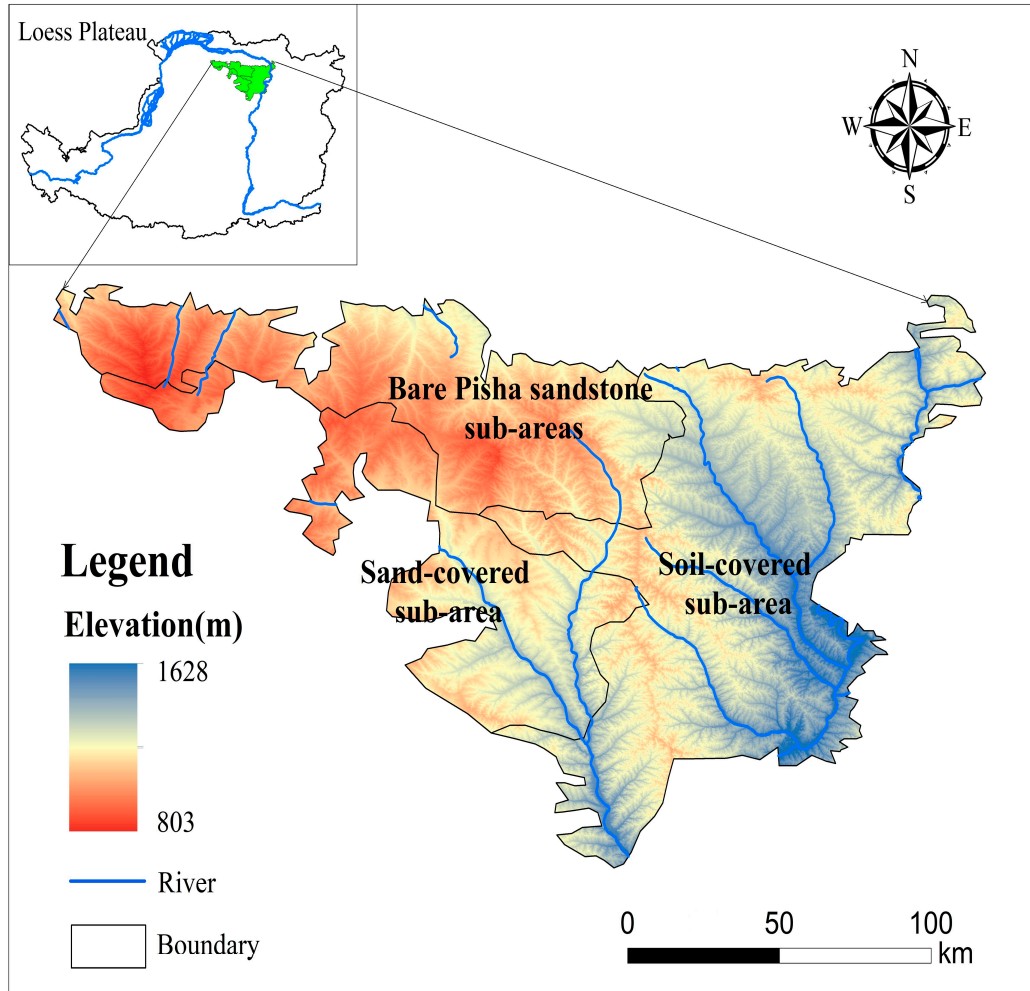

**Figure 1.** Location of the study area.

The precipitation and temperature data were obtained from the National Earth System Science Data Center (http://loess.geodata.cn, on 13 September 2020), with a spatiotemporal resolution of 1000 m and month from 1998 to 2017, respectively. The annual precipitation data were obtained by summing the monthly precipitation data, and the annual average temperature data were obtained by averaging the monthly average temperature data. The data were verified using data from 496 independent meteorological observation points nationwide in China, and the verification results were credible. NDVI was provided by the website of the Institute of Geographical Sciences and Natural Resources Research, Chinese Academy of Sciences (http://www.resdc.cn, accessed on 1 September 2019), with a spatiotemporal resolution of 1000 m, monthly and annually, from 1998 to 2017, respectively. Land use data from 1998 to 2017 were obtained from the products provided by Yang and Huang, with a 30 m spatial resolution [43].

### 2.2. Methodology

2.2.1. Calculate Vegetation Coverage

An evaluation framework has been established to identify the variation of VEC and quantitatively assess the impact of environmental factors on its spatial distribution in the PSA, as shown in Figure 2. The framework describes each part of the study and the corresponding analytical methods used in detail. In many previous studies, VEC is usually calculated using NDVI data. In this study, VEC is also calculated based on NDVI, and the calculation formula is as follows [30,44]:

$$f_i = (NDVI_i - NDVI_{min})/(NDVI_{max} - NDVI_{min}) \tag{1}$$

where $f_i$ is the VEC of the *i*-th cell; $NDVI_i$ is the *NDVI* value of the *i*-th cell; $NDVI_{max}$ is the maximum *NDVI* value of the entire study area; $NDVI_{min}$ is the minimum *NDVI* value of the entire study area.

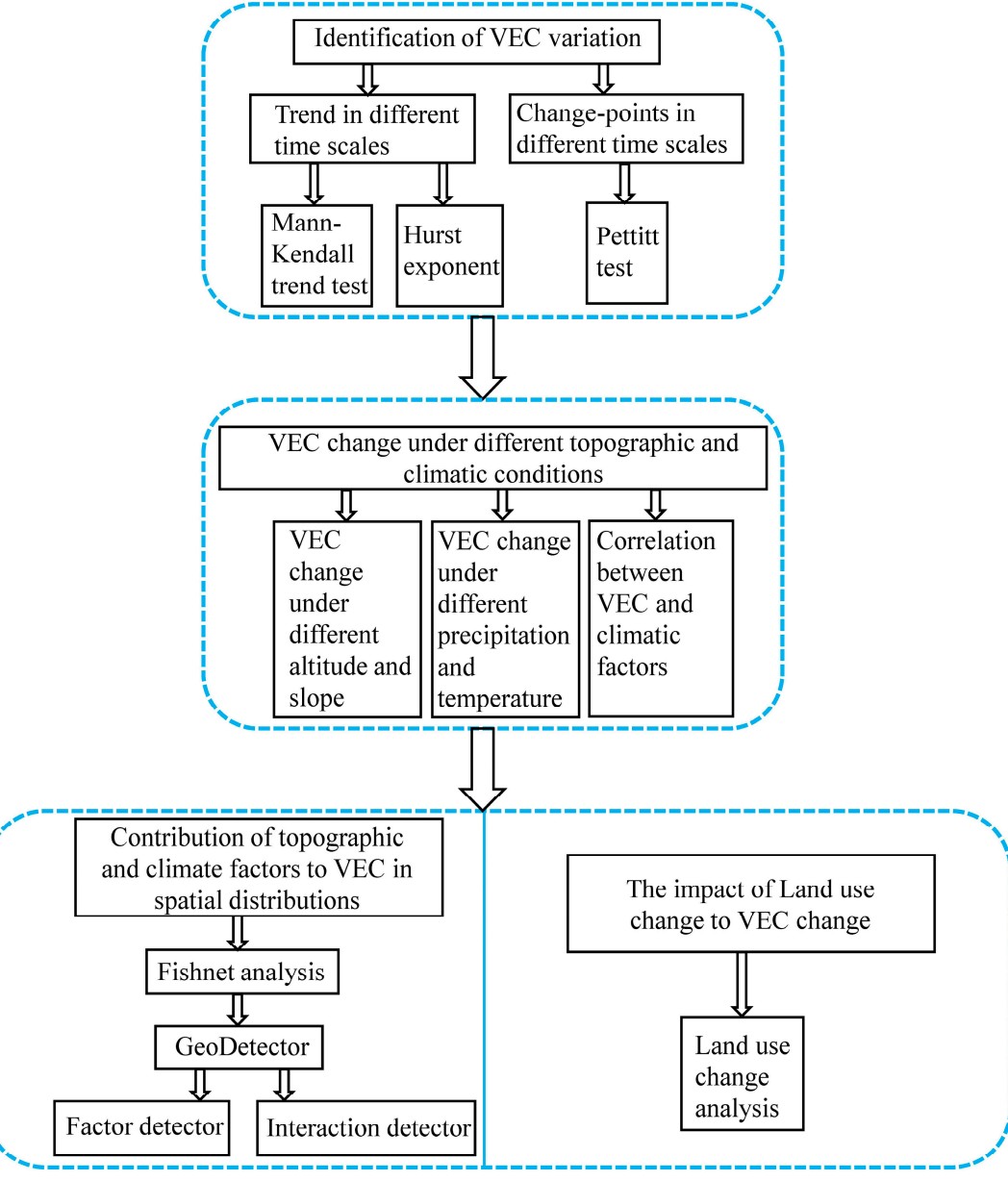

**Figure 2.** Assessment Framework.

### 2.2.2. Mann–Kendall (MK) Trend Test Method

The Mann–Kendall (MK) trend test method is widely used in the fields of meteorology and hydrology [45,46]. This study used the MK trend test method to analyze the changing trend of VEC and climatic factors. The statistic $S$ is calculated as follows:

$$S = \sum_{i=1}^{n-1} \sum_{j=i+1}^{n} \text{sgn}(x_j - x_i) \tag{2}$$

where $x$ is the variable to be analyzed; $n$ is the number of samples of the variable; the calculation formula of the function sgn $(x_j - x_i)$ is as follows:

$$\text{sgn}(x_j - x_i) = \begin{cases} +1, & x_j - x_i > 0 \\ 0, & x_j - x_i = 0 \\ -1, & x_j - x_i < 0 \end{cases} \tag{3}$$

The formula for calculating the variance of the statistic $S$ is as follows:

$$Var(S) = \frac{n(n-1)(2n+5)}{18} \tag{4}$$

When $n > 10$, the standard normalization statistic $Z$ could be represented as:

$$Z = \begin{cases} \frac{S-1}{\sqrt{Var(S)}}, & S > 0 \\ 0, & S = 0 \\ \frac{S+1}{\sqrt{Var(S)}}, & S < 0 \end{cases} \tag{5}$$

When $Z$ is a positive value, it indicates that the change of the variable is an increasing trend; when $Z$ is a negative value, it indicates that the change of the variable is a decreasing trend; when the absolute value of $Z$ is greater than 1.96, it indicates that the change trend is significant at the level 0.05.

### 2.2.3. Hurst Exponent

Hurst exponents are methods for analyzing the persistence of time variables in various fields [30,44,47]. Furthermore, this method is often used in research on VEC. The main formula is as follows:

$$\frac{R(\tau)}{S(\tau)} = (a\tau)^H \tag{6}$$

where $R(\tau)$ is the range, $S(\tau)$ is the standard deviation of the range, $a$ is a constant, $\tau$ is the time period, and $H$ is the Hurst exponent. $H$ is obtained by fitting the equation $\log(R/S)\tau = a + H \times \log(\tau)$.

The range of $H$ value is 0–1. When $H > 0.5$, it indicates that the change of the time series is consistent with the past; when $H < 0.5$, it indicates that the change of the time series is opposite to the past; when $H = 0.5$, it indicates time series is random.

### 2.2.4. Pettitt Test

The Pettitt test is used to identify the change points of VEC [48]. The Pettitt test method can be described as follows:

$$p = 2e^{\left(\frac{-6U_{t,n}^2}{n^2+n^3}\right)} \tag{7}$$

where $n$ is the length of time for vegetation coverage data; $p$ is the significance level. When the $p$-value is less than 0.05, it is considered that there is a significant change-point; $Ut, n$ are statistics constructed according to vegetation coverage.

### 2.2.5. GeoDetector Based on Fishnet Analysis

GeoDetector can detect the reason for the change of variables in spatial distribution [36,49]. This study applies factor detectors and interaction detectors in GeoDetector developed by Song et al. to quantitatively evaluate vegetation cover driving forces and multi-factor interactions [50]. The interaction between multi-factor is divided into five types, namely bi-variable enhance, nonlinear-enhance, uni-variable weaken, nonlinear-weaken, and independent. The factor detector is used to detect the influence of various factors on the spatial differentiation of VEC. The higher the $q$ value, the greater the influence. The expression is:

$$q = 1 - \frac{\sum\limits_{h=1}^{L} N_h \sigma_h^2}{N \sigma^2} = 1 - \frac{SSW}{SST} \tag{8}$$

$$SSW = \sum_{h=1}^{L} N_h \sigma_h^2 \tag{9}$$

where $N$ is the number of all samples in the study area, $\sigma^2$ is the variance of the VEC in the study area, $h$ is the stratum, and $L$ represents the number of stratums. $N_h$ and $\sigma_h^2$ are the number of all samples and the variance of VEC in the $h$-th layer, respectively. The value range of $q$ is 0–1. The closer it is to 1, the greater the impact of the factor on the distribution of VEC.

Based on the fishnet analysis tool in ArcGIS, the study area was divided into 266 grids with an 8000 m spatial resolution (Figure 3), and each grid includes at most 64 sample cells of VEC. Using GeoDetector, the influence factors of VEC in each grid are quantitatively assessed. This way, the spatial distribution of the impact of different factors on VEC can be obtained.

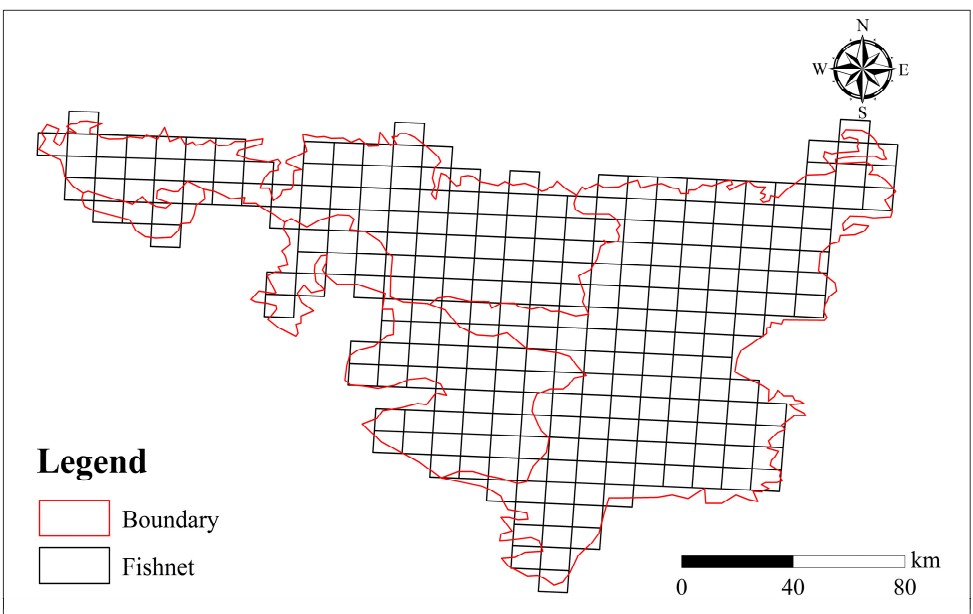

**Figure 3.** Spatial distribution of fishnet in the study area.

## 3. Results

### 3.1. Trend of VEC

The multi-year average annual VEC in the PSA, the SASA, the SOSA, and the BSA were 0.42, 0.41, 0.49, and 0.30, respectively. The percentage area of a significant increase in annual VEC reached 63.89% in the PSA ($p < 0.05$), which was distributed in the SASA, the SOSA, as well as the east and west of the BSA (Figure 4a). The average Hurst exponent of the annual VEC in the SASA, the SOSA, the BSA, and the PSA were 0.63, 0.61, 0.59, and

0.61, respectively. The area with a Hurst exponent greater than 0.5 accounted for 92.66%, where the annual VEC presented positive continuity (Figure 4b). The region where VEC decreased significantly ($p < 0.05$) in February and December was larger, reaching 14.92% and 13.75%, respectively, and was distributed in the BSA and the southern part of the SASA. From April to November, the area where VEC significantly increased exceeded 30% of the total area (Figure 5). Except for January, in the remaining months, VEC was positively continuous in more than 60% of the area, indicating that the overall situation of vegetation restoration in this area was good (Figure 5).

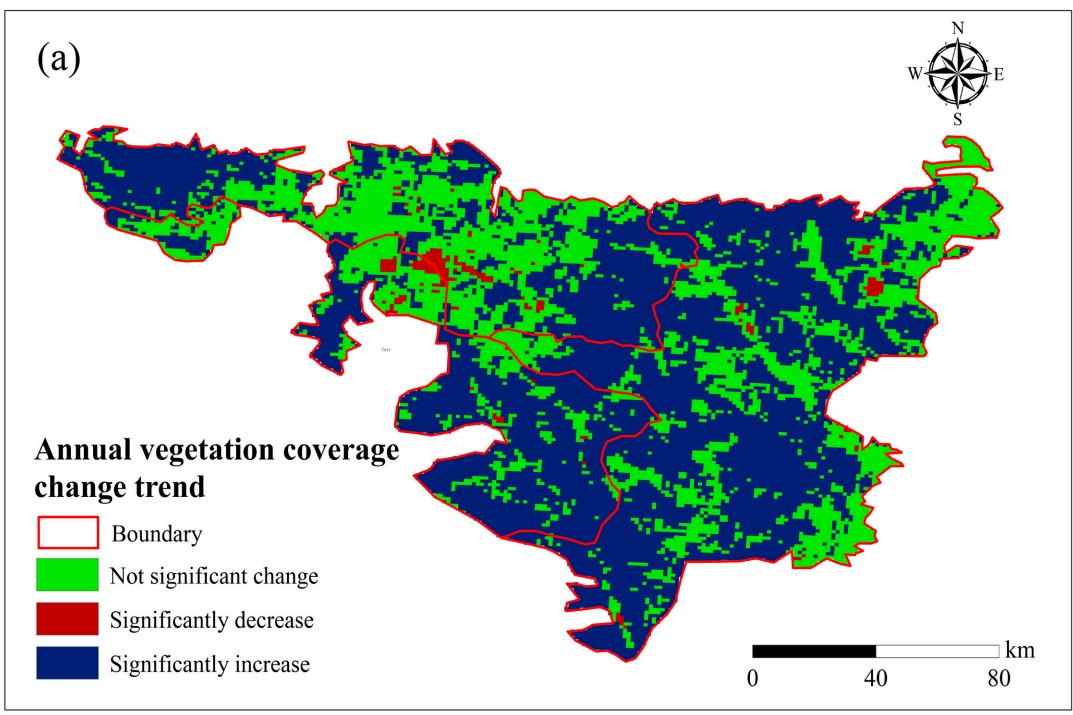

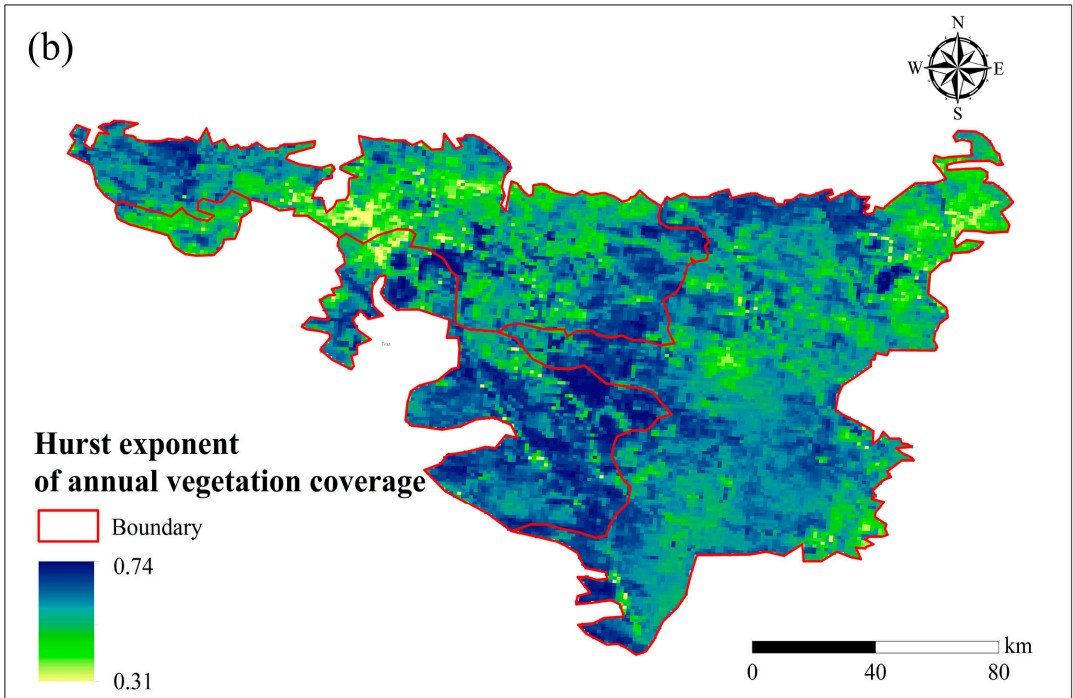

**Figure 4.** Spatial distribution of trend (**a**) and Hurst exponent (**b**) of annual VEC.

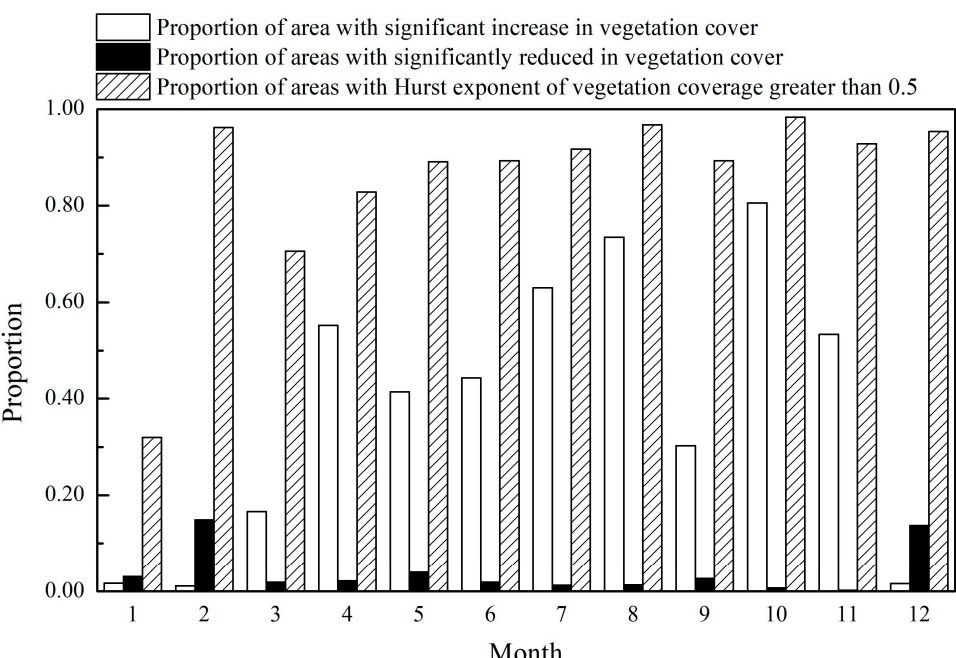

**Figure 5.** Trend characteristics of VEC in different months.

### 3.2. Identification of Significant Change Points in VEC

According to Figure 6, significant change points in annual VEC primarily occurred between 2002 and 2012. The percentages of the total area with these change points in the PSA, the SASA, the SOSA, and the BSA, were 52.45%, 13.69%, 26.75%, and 12.01%, respectively. These change points were primarily concentrated in the south of the SASA, the west of the SOSA, and the east and the west of the BSA ($p < 0.05$). Moreover, there were significant change points in VEC for each month ($p < 0.05$), but there was no consistent pattern in their spatial distribution (as shown in Figure 7). From January to December, the percentages of the total area with significant change-points of VEC were 1.96%, 9.03%, 8.30%, 39.05%, 37.00%, 36.65%, 53.68%, 63.02%, 24.47%, 66.88%, 18.78%, and 13.65%, respectively. As the months progressed, the area with significant change points in VEC gradually increased, reaching its maximum in October. This confirmed that the increased strength in VEC during this period may be the largest.

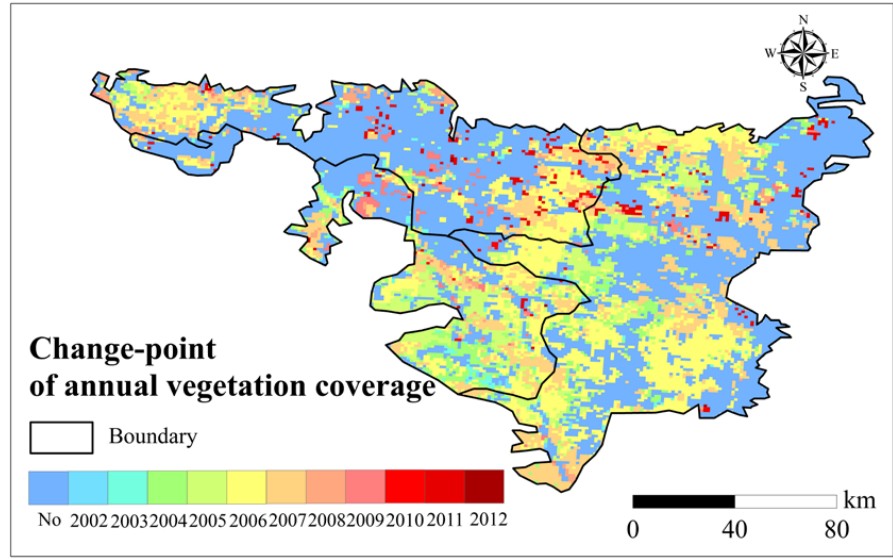

**Figure 6.** Spatial distribution of change points of annual VEC.

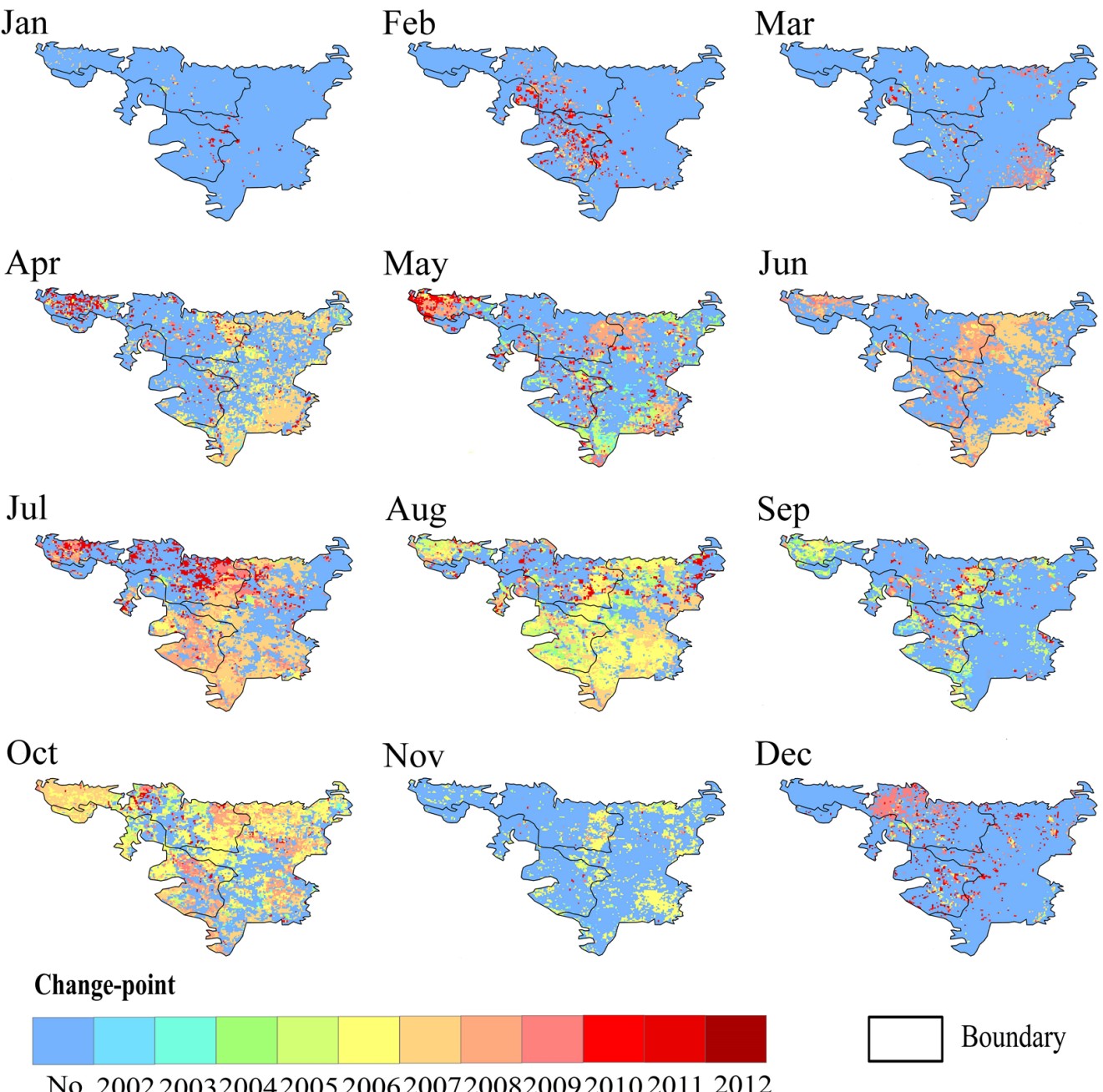

**Figure 7.** Spatial distribution of significant change points of monthly VEC.

### 3.3. VEC Changes under Different Topographic Conditions

Topographic factors can alter the spatial distribution of water and heat, indirectly affecting vegetation growth. At most monthly and annual scales, the percentage of the area where VEC had significantly increased was mainly in the altitude range of 1050–1500 m or the slope range of 0–21° ($p < 0.05$) (Figure 8a,b). The altitude range and slope range of the areas where VEC had significant change points were consistent with the areas where VEC significantly increased ($p < 0.05$) (Figure 8c,d). This suggests that vegetation is more likely to thrive in the middle to low altitude and lower slope ranges.

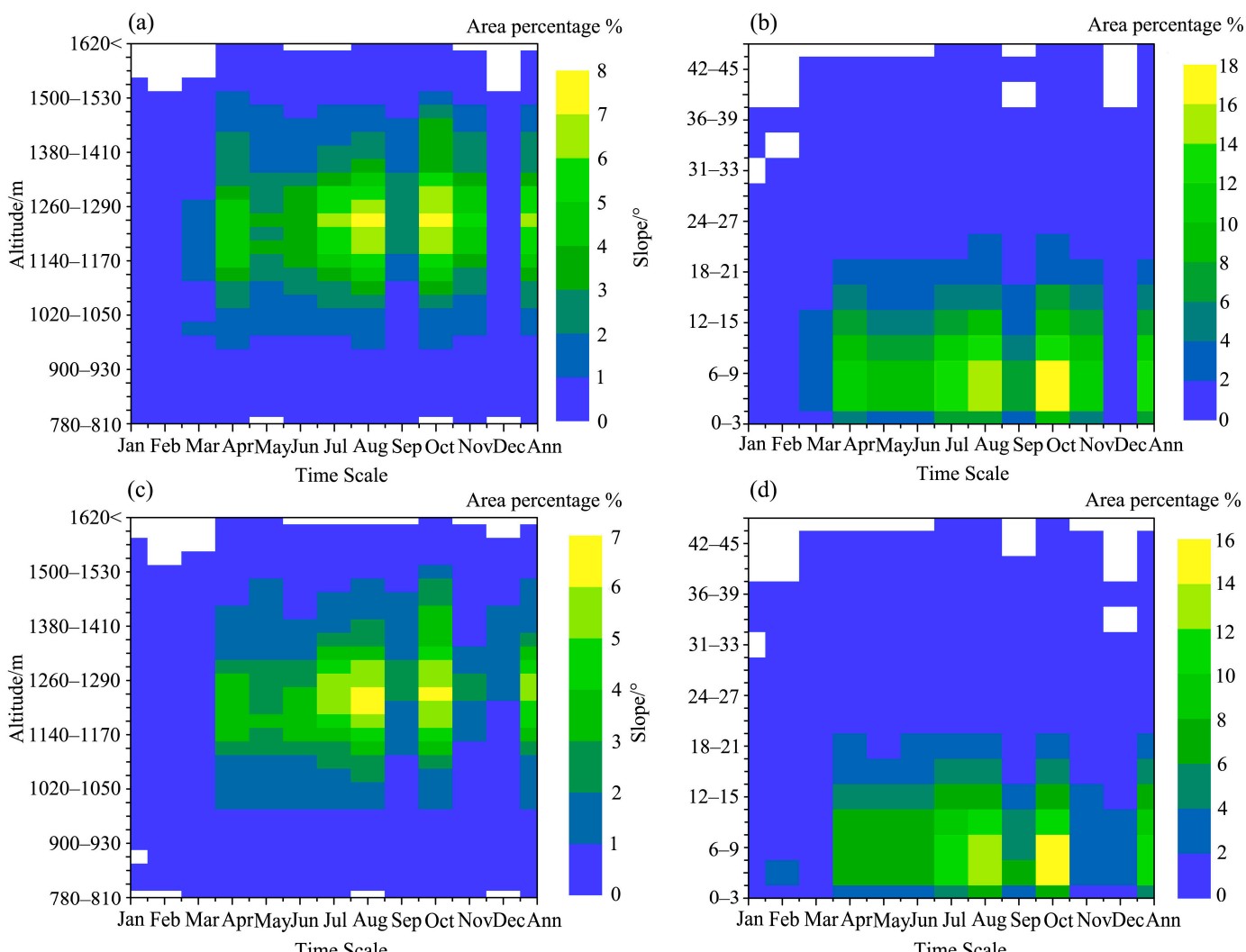

**Figure 8.** In different altitude ranges and slope ranges, the percentage of areas where VEC had a significant increasing trend at different time scales (**a**,**b**); the percentage of areas where VEC had significant change-points (**c**,**d**).

As the altitude increased, annual VEC decreased, while annual VEC gradually increased as the slope increased (Figure 9a,b). Figure 9c shows that for different years, the most suitable range for VEC during the flood season was mainly at the lower altitude range, whereas during the non-flood season, VEC was higher in high-altitude ranges. Except for a few individual months of certain years, VEC was higher in lower slope ranges. VEC was higher in high slope ranges for most periods (Figure 9d).

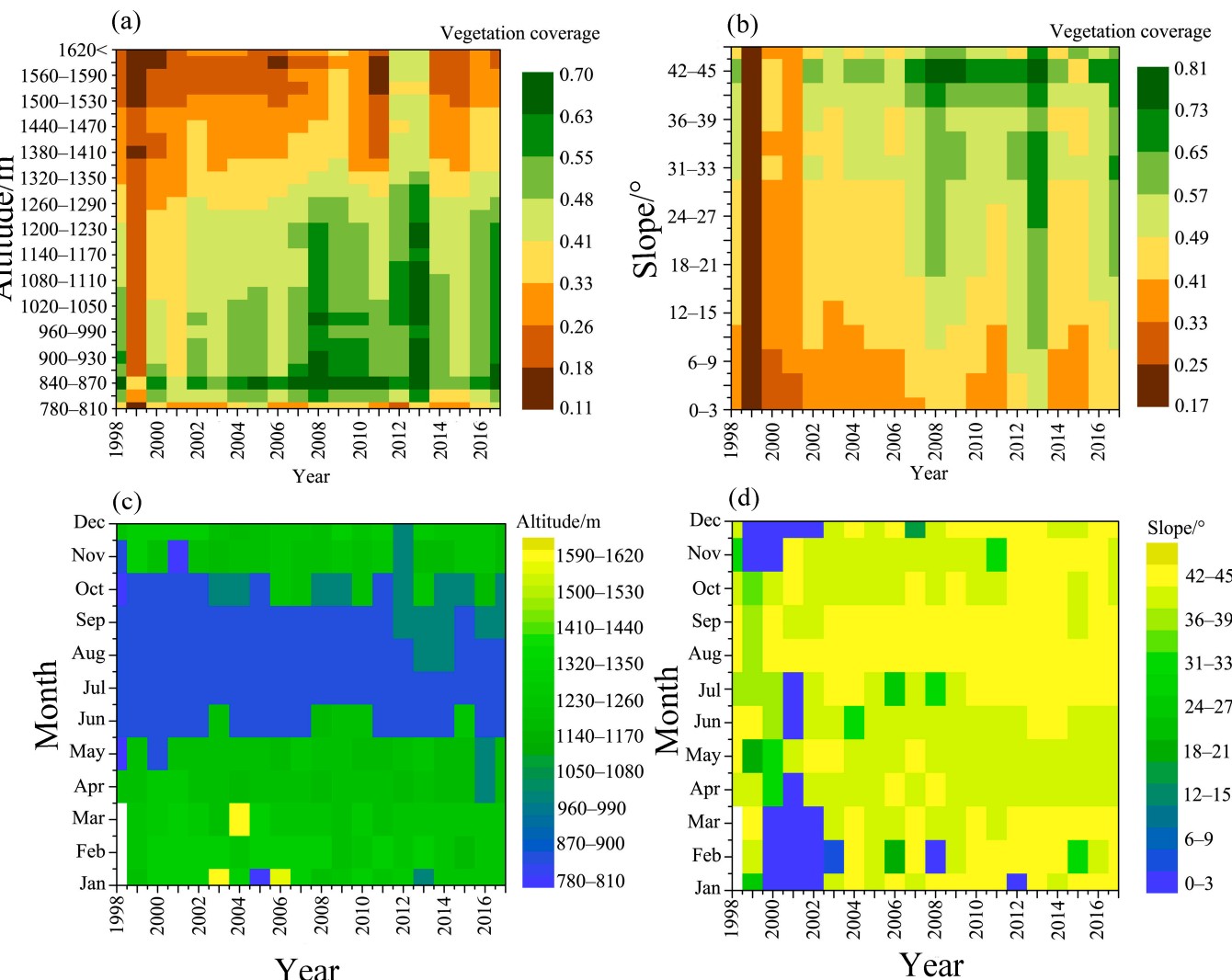

**Figure 9.** Average annual VEC in different altitude ranges (**a**); average annual VEC in different slope ranges (**b**); the altitude range with the highest average VEC in different months each year (**c**); the slope range with the highest average VEC in different months each year (**d**).

### 3.4. VEC Change under Different Climatic Conditions

From 1998 to 2017, a significant increasing trend occurred in annual precipitation in the south of the PSA ($p < 0.05$) (Figure 10a), while the trend of annual average temperature increase was not significant in the PSA ($p > 0.05$) (Figure 10b). In the PSA, the percentage of areas where annual VEC was significantly correlated with annual precipitation reached 45.35% ($p < 0.05$) (Figure 10e), distributed in the eastern part of the BSA and the SOSA. This confirmed that the increase in VEC may be due to the effects of precipitation on an annual scale.

Especially in July and October, there was a significant positive correlation between precipitation and VEC on a large scale, with the area accounting for 64.99% and 39.21%, respectively ($p < 0.05$). Multi-year average precipitation and multi-year average annual average temperature increased gradually from northwest to southeast (Figure 10c,d). The areas with high multi-year average annual precipitation were consistent with the areas where the annual VEC and annual precipitation were significantly positively correlated ($p < 0.05$). Overall, 62.18% of the area had an insignificant negative correlation between annual VEC and annual average temperature ($p > 0.05$) (Figure 10f).

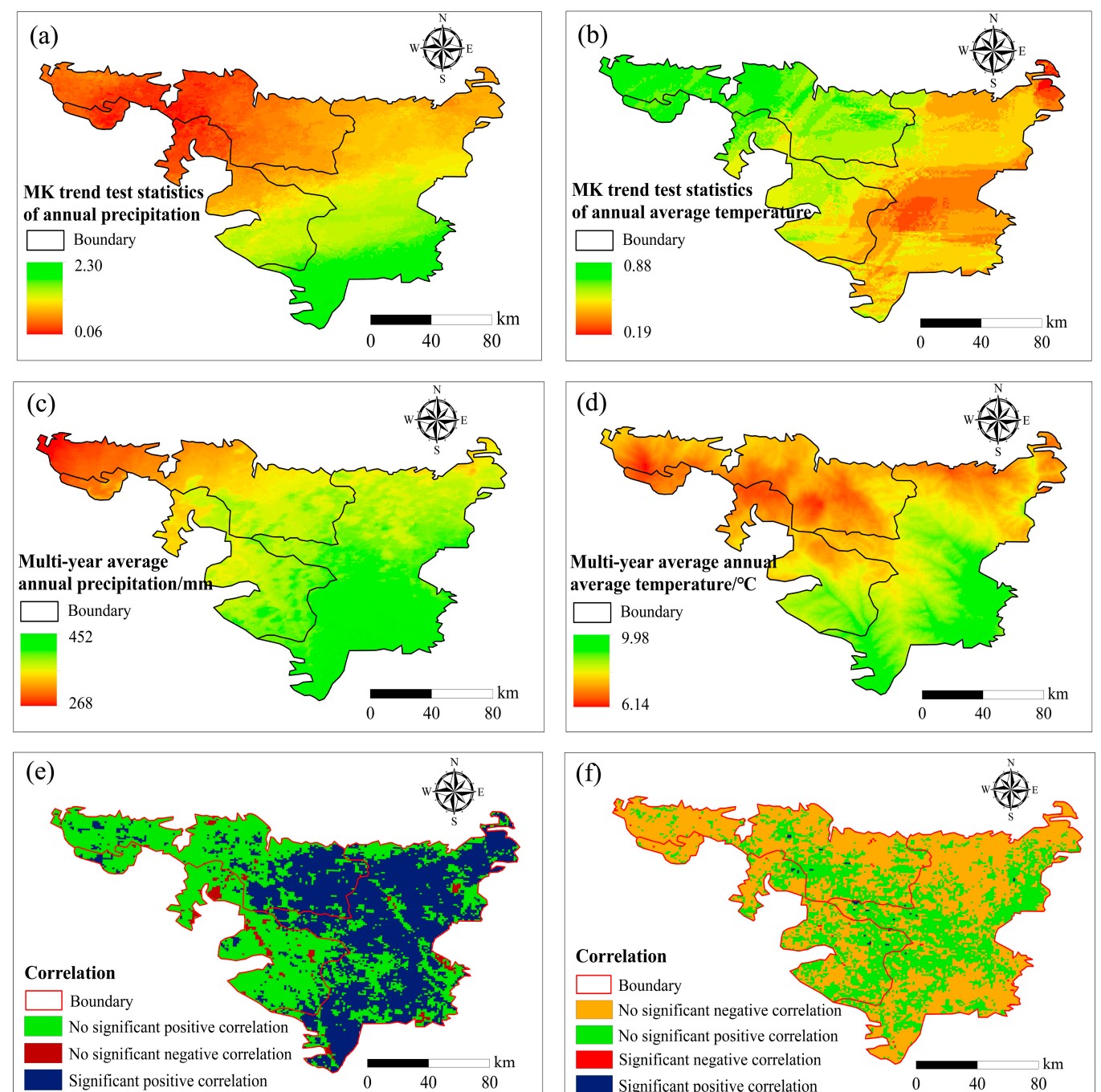

**Figure 10.** Spatial distribution of trend of annual precipitation (**a**); spatial distribution of trend of annual average temperature (**b**); spatial distribution of multi-year average annual precipitation (**c**); spatial distribution of multi-year average annual temperature (**d**); spatial distribution of correlation between annual precipitation and annual VEC (**e**); spatial distribution of correlation between annual average temperature and annual VEC (**f**).

When precipitation and the annual average temperature increased, respectively, VEC also increased annually (Figure 11a,b). The highest value of VEC was mainly distributed in the range with the highest precipitation at annual and monthly scales (Figure 11a,c), while on an annual scale, the highest value of VEC was mainly distributed in the average temperature range of 8–9 °C or 9–10 °C (Figure 11b). For different years, the precipitation range of the highest value distribution of VEC during the flood season was higher, and the temperature range of the highest value of VEC in July was the highest (Figure 11d).

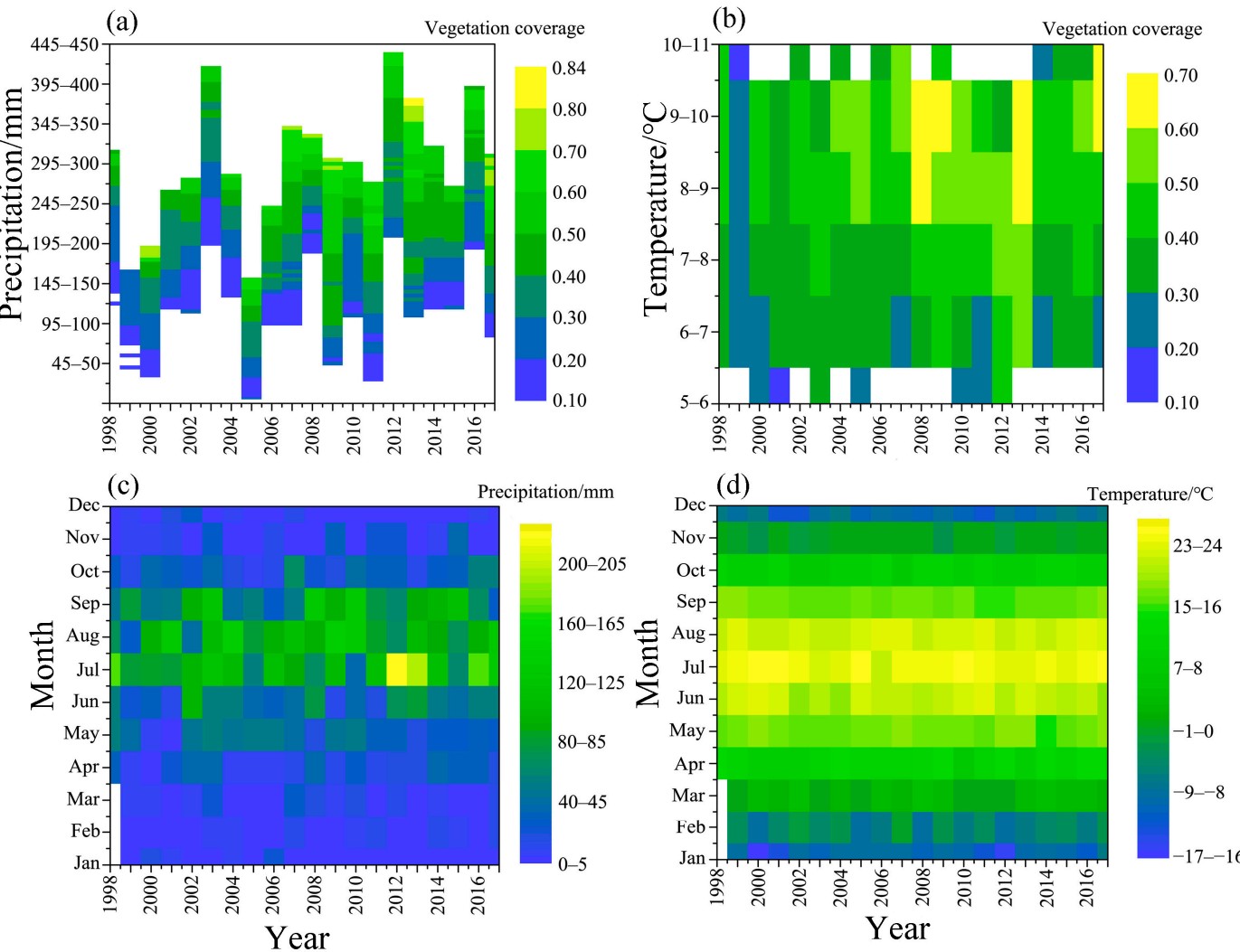

**Figure 11.** Average annual VEC in different annual precipitation ranges (**a**); average annual VEC in different annual average temperature ranges (**b**); the precipitation range with the highest average VEC in different months each year (**c**); the average temperature range with the highest average VEC in different months each year (**d**).

## 4. Discussion

### 4.1. Contribution of Topographic and Climate Factors to VEC in Spatial Distributions

VEC changes were affected by many factors simultaneously, including topographic factors such as slope and altitude [19,51]. The research showed that when the slope was less than 35°, it was indeed beneficial to vegetation, while the VEC increased the most in the slope range of 15–25° in China [30,44]. Therefore, in arid regions with extreme water shortages, topographic factors were crucial to VEC changes [52]. Moreover, climate factors were key factors affecting terrestrial ecosystems, including VEC changes [53,54]. Especially in arid regions, vegetation was more sensitive to precipitation because lack of water limits the growth of vegetation [55,56]. Excessively high temperature may increase water evaporation in some areas and is not conducive to VEC growth.

To further quantitatively assess the contribution of topographic and climatic factors on VEC, the contribution of the multi-year average precipitation, multi-year average temperature, elevation, and slope to the multi-year average VEC of 266 grids in the study area was analyzed using the factor detector of the GeoDetector (Figure 12). The areas where the four factors contributed less to the VEC were mainly distributed in the central area, and the areas with greater contributions were scattered. This indicated that there were great

spatial differences in the impact of various factors on VEC, and the impact of a single factor on VEC may be limited.

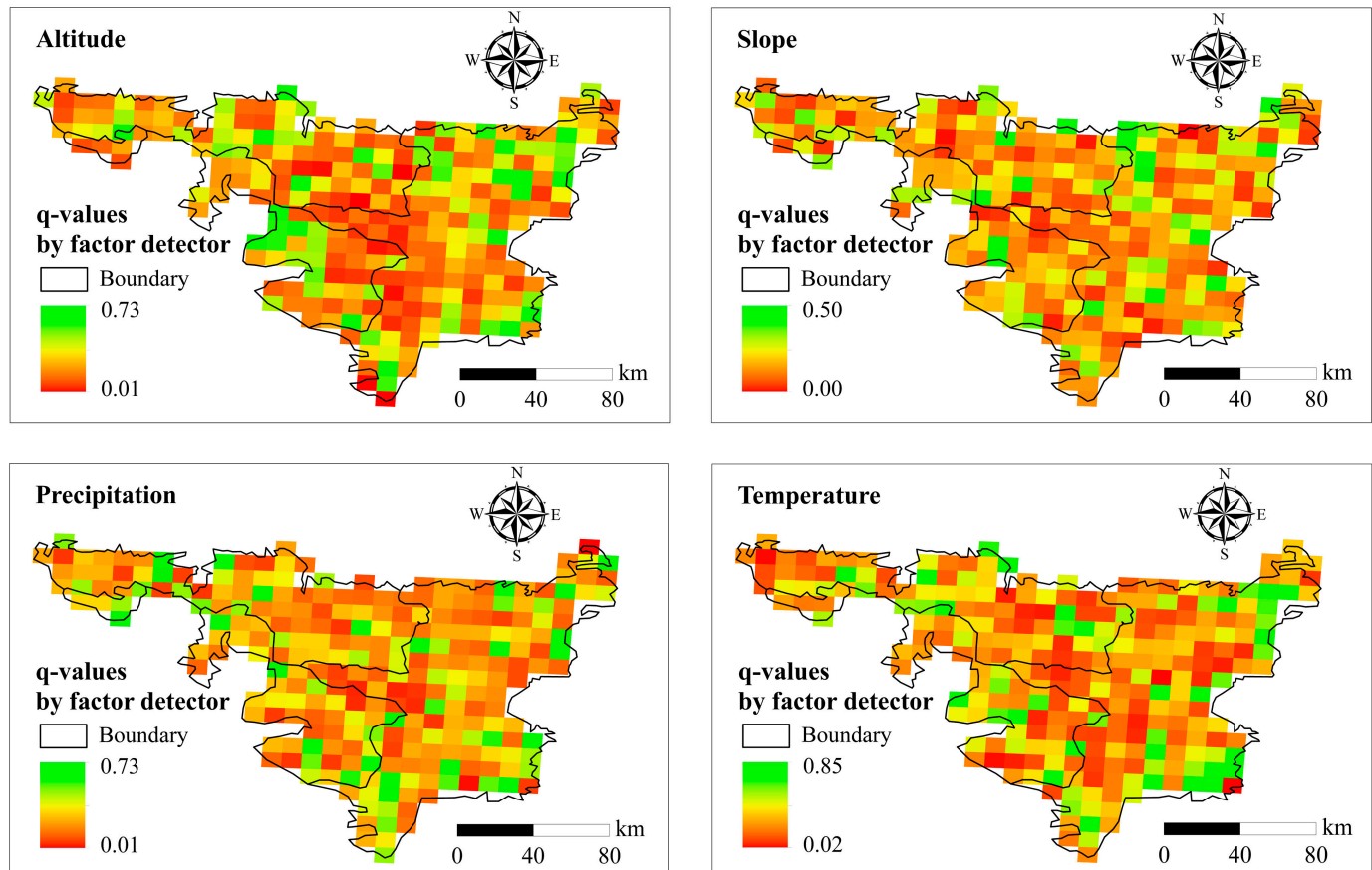

**Figure 12.** Spatial distribution of q-values by factor detector in fishnet.

Using the interactive detector of GeoDetector, the interactive relationship between different influencing factors on VEC was analyzed (Figure 13). In most areas, the interaction relationship between each pair of factors was enhanced. The nonlinear enhancement of the interaction between altitude and slope on VEC showed an obvious clustering distribution in the central part of the PSA. The nonlinear enhancement of the interaction between temperature and slope on VEC also exhibited a similar pattern. This confirmed that VEC was not linearly affected by a single factor in most PSA. The study by Ma et al. also confirmed that precipitation, soil moisture, and air temperature affected VEC changes in the PSA [51].

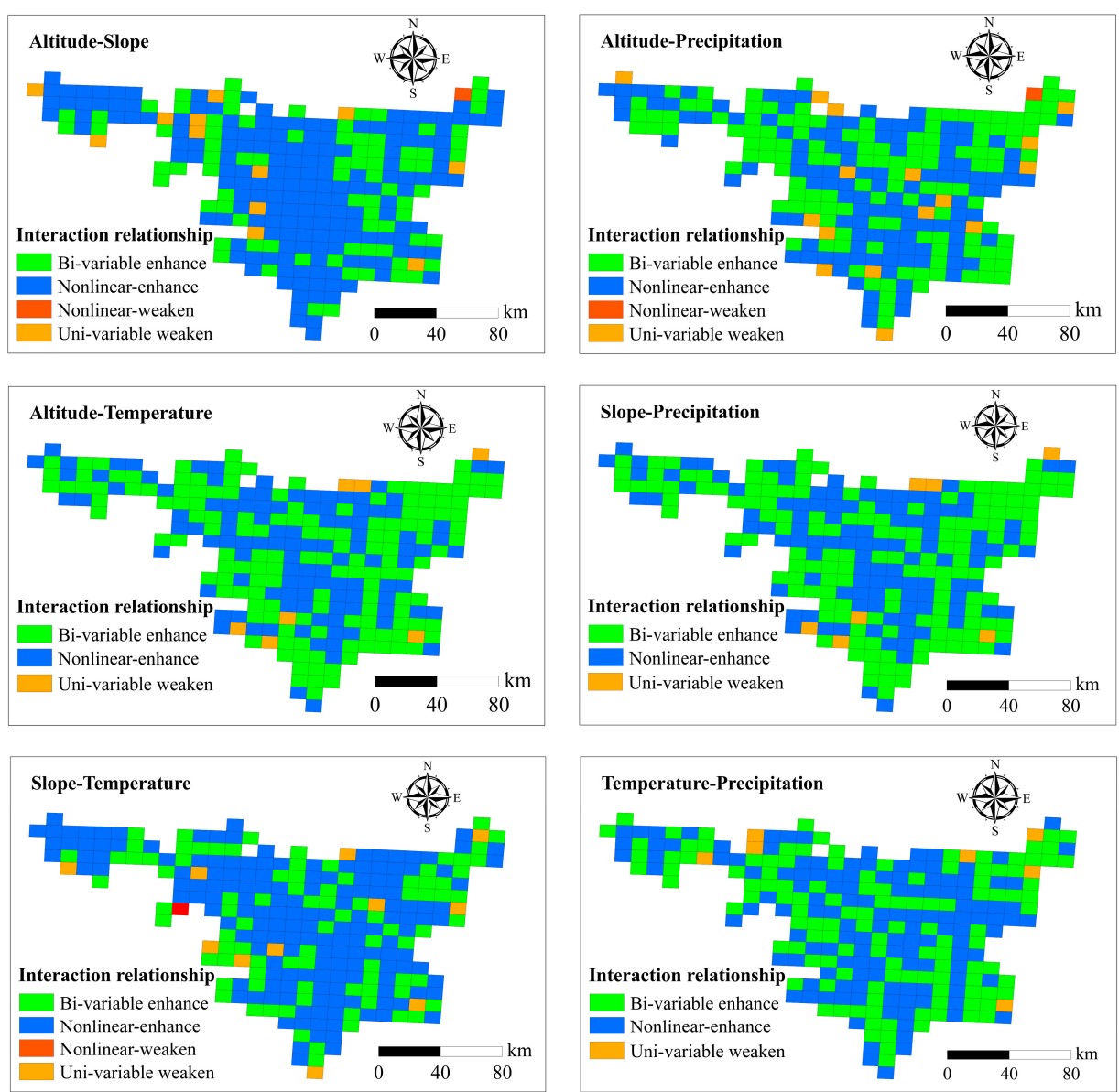

**Figure 13.** Spatial distribution of interactive relationship between different factors in fishnet.

### 4.2. The Impact of Land Use Change on VEC Change

In addition to topographic factors and climatic factors, human activities also affect VEC [57,58]. Sun et al. showed that land use played a crucial role in NDVI in some regions [59]. The Loess Plateau underwent significant changes in land use after decades of returning farmland to forests, with planted forest areas increasing significantly from 1998 to 2006 [60].

Grassland area accounted for 78.75% and 87.46% of the PSA in 1998 and 2017, respectively, while cropland area accounted for 15.99% and 8.73%. Du et al. also confirmed that grassland and cropland were the main types of land use in the PSA, and land use changes played a role in reducing wind erosion [40,61]. Compared with 1998, the forest, grassland, water, and impervious in 2017 increased obviously, with the percentages of increase being 0.06%, 8.89%, 0.04%, and 1.50% in the PSA, respectively (Table 1). The grassland area increased the most in the SOSA. Compared with 1998, the cropland and barren in 2017 decreased, confirming that returning farmland to forests had effectively increased VEC, especially in the SOSA. However, the effect was limited compared to the area where VEC was significantly increased in the PSA. Therefore, climatic factors, especially the increase in

precipitation, may lead to a VEC increase in a large area. Ma et al. analyzed the impact of land use on VEC in the region, indicating that land use change had a small impact on VEC change, further confirming the speculation in this study.

**Table 1.** Changes in the proportions of various types of land use in the PSA and its sub-regions in 2017 compared to 1998.

| Land Use | Rate of Change in the Entire PSA | Rate of Change in the BSA | Rate of Change in the SASA | Rate of Change in the SOSA |
|---|---|---|---|---|
| **Cropland** | **−7.26%** | **−0.85%** | **−0.74%** | −5.68% |
| Forest | 0.06% | 0.01% | 0.03% | 0.03% |
| Grassland | 8.89% | 2.12% | 1.39% | 5.37% |
| Water | 0.04% | 0.01% | 0.03% | 0.01% |
| Barren | −3.23% | −1.47% | −1.20% | −0.56% |
| Impervious | 1.50% | 0.18% | 0.49% | 0.83% |

Vegetation growth in arid and semi-arid regions was severely restricted by water conditions [62]. A reasonable vegetation restoration strategy should be formulated based on hydrological conditions and other environmental factors, and appropriate drought-tolerant tree species should be selected for planting in different spatial distributions [60,63].

**5. Conclusions**

Research on the VEC is meaningful for understanding ecological quality and soil erosion in the PSA, which is an area in the Loess Plateau with an extremely poor ecological environment. On the one hand, the annual VEC in 63.89% of the area showed a significant increase in the PSA from 1998 to 2017 ($p < 0.05$). In various months of the year, particularly from April to November, the VEC showed a significant increase in areas exceeding 30% of the total area ($p < 0.05$). The VEC would be expected to continue to increase in the future, as predicted by the Hurst exponent. At different time scales, significant change points of VEC were observed between 2002 and 2012 ($p < 0.05$). On the other hand, the greater the precipitation, the more favorable the VEC would increase at different scales. Compared with temperature, precipitation had a greater impact on VEC. There was a significant positive correlation between annual VEC and annual precipitation in 45.35% of the area ($p < 0.05$). The altitude range of 1050–1500 m was the most conducive to VEC, and the slope range of 0–21° was the slope range that was most suitable for VEC. In most of the PSA, the enhanced interaction relationship between different factors on VEC was ubiquitous. In the future, the management and maintenance of vegetation restoration should be carried out more reasonably following the natural environment in the PSA.

**Author Contributions:** Conceptualization, K.Y. and H.L.; Methodology, L.J.; Software, K.Y.; Validation, K.Y. and Z.L.; Formal analysis, Z.L.; Resources, Z.R.; Data curation, Z.R.; Writing—original draft, L.J.; Writing—review & editing, P.L. All authors have read and agreed to the published version of the manuscript.

**Funding:** This research was supported by the National Key Research and Development Program of China (No. 2022YFF1300800) and the National Natural Science Foundation of China (No. 52022081), and the Key Research and Development Program of Shaanxi Province (2023-ZDLSF-60).

**Institutional Review Board Statement:** Not applicable.

**Informed Consent Statement:** Not applicable.

**Data Availability Statement:** Not applicable.

**Conflicts of Interest:** The authors declare no conflict of interest.

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
