# Peer review of "Identification of Vegetation Coverage Variation and Quantitative the Impact of Environmental Factors on Its Spatial Distribution in the Pisha Sandstone Area"

_sustainability, doi:10.3390/su15076054_

Round 1

Reviewer 1 Report

Taking the Loess Plateau as an example, the authors analyzed the identification of vegetation cover change and the quantitative impact of environmental factors on its spatial distribution in Pisha Sandstone area. This is a meaningful research for ecological restoration. However, the manuscript still has major problems. Especially, the structure and format are chaotic. I suggest that authors carefully revise and resubmit this manuscript in the journal.

Major comments:

1Please enhance the academic of the language expression of the manuscript.

2The structure of the paper is unreasonable. Materials and Methods should be the same part.  The data analysis section is combined with the results section, which is uncommon in a scientific paper.

3Introduction: This is not sufficient. There is less research background and importance. At the same time, the practical significance of the study is not obvious.

4Study area: There are too few introductions about the study area. Readers can't understand the location of the study area well. You can briefly introduce the precipitation, climate, temperature and vegetation coverage of these areas.

5Flawed methodology: The authors directly shows the formula, and introduces the method itself less. It is difficult for readers to understand the meaning of these methods, especially for readers like me who are not very good at this aspect.

6Discussion: 5.2.The impact of the implementation of the policy of returning farmland to forest. What is the relationship between this part and the previous results. It is unreasonable to use only one policy to analyze the impact of human activities on vegetation coverage. Human activities such as population growth and economic development will have an impact on -vegetation coverage. The discussion part should analyze the research results, such as comparing the differences between this study and other studies. This part does not seem to be the result of the study. What if it is reasonable to discuss the impact of policy?

Specific comments:

Lines 12-13: PSAVEC: What does this mean? Please use the full name when it first appears.

Lines 44: the research shows should be replaced with researches show.

Lines 82: km2 should be replaced with hm2

Lines 89: was should be replaced with is

Lines 97-100: What is NDVI? The authors should describe in detail.

Reviewer 3 Report

The issues discussed in this paper are really relevant and of scientific interest due to the presence of a number of insufficiently considered individual aspects and some other reasons. Identification of vegetation coverage variation and quantitative the impact of environmental factors on its spatial distribution can also be singled out among them. The scientific novelty and practical importance of this work is due to the calculations performed.  In the Introduction chapter, it can be seen that the authors gave an overview of the current situation, but there is still a lot of literature that is not included in the list, which would still give a more precise picture of a better understanding of the the mentioned issue.  The material and method of the chapters are well structured, but they need to be revised and the literature used expanded to be acceptable to the readers. In the paper should also include a review/comparison with more other authors who have dealt with similar experiments and results.

Round 2

Reviewer 1 Report

The manuscript has been nicely improved after revision. But the language still needs polishing.
